# Muscle systems and motility of early animals highlighted by cnidarians from the basal Cambrian

Xing Wang[1,2]*[†], Jean Vannier[3][†], Xiaoguang Yang[4][†], Lucas Leclère[5], Qiang Ou[6], Xikun Song[7], Tsuyoshi Komiya[8], Jian Han[4]*

[1]Qingdao Institute of Marine Geology, China Geological Survey, Qingdao, China; [2]Function Laboratory of Marine Mineral Resources, Qingdao National Laboratory for Marine Science & Technology, Qingdao, China; [3]Université de Lyon, Université Claude Bernard Lyon 1, CNRS UMR 5276, Laboratoire de géologie de Lyon: Terre, Planètes, Environnement, Bâtiment GEODE, Villeurbanne, France; [4]State Key Laboratory of Continental Dynamics, Shaanxi Key laboratory of Early Life & Environments, Department of Geology, Northwest University, Xi'an, China; [5]Sorbonne Université, CNRS, Laboratoire de Biologie du Développement de Villefranche-sur-Mer (LBDV), Villefranche-sur-mer, France; [6]Early Life Evolution Laboratory, School of Earth Sciences & Resources, China University of Geosciences, Beijing, China; [7]State Key Laboratory of Marine Environmental Science, College of Ocean and Earth Sciences, Xiamen University, Xiamen, China; [8]Department of Earth Science & Astronomy, Graduate School of Arts & Sciences, The University of Tokyo, Tokyo, Japan

**\*For correspondence:**
wx5432813@126.com (XW);
elihanj@nwu.edu.cn (JH)

[†]These authors contributed equally to this work

**Competing interest:** The authors declare that no competing interests exist.

**Abstract** Although fossil evidence suggests that various animal groups were able to move actively through their environment in the early stages of their evolution, virtually no direct information is available on the nature of their muscle systems. The origin of jellyfish swimming, for example, is of great interest to biologists. Exceptionally preserved muscles are described here in benthic peridermal olivooid medusozoans from the basal Cambrian of China (Kuanchuanpu Formation, ca. 535 Ma) that have direct equivalent in modern medusozoans. They consist of circular fibers distributed over the bell surface (subumbrella) and most probably have a myoepithelial origin. This is the oldest record of a muscle system in cnidarians and more generally in animals. This basic system was probably co-opted by early Cambrian jellyfish to develop capacities for jet-propelled swimming within the water column. Additional lines of fossil evidence obtained from ecdysozoans (worms and panarthropods) show that the muscle systems of early animals underwent a rapid diversification through the early Cambrian and increased their capacity to colonize a wide range of habitats both within the water column and sediment at a critical time of their evolutionary radiation.

## Editor's evaluation

Based on exceptionally preserved fossils of olivooids, a group of early cnidarians, from the basal Cambrian of China (535 million years ago), Wang and colleagues reveal primitive muscles of early animals with well-developed system of circular fibers directly comparable with the myoepithelial muscles of modern medusae, representing the oldest record of a muscle system in cnidarians and more generally in animals. The manuscript will be of broad interest to scientists, including paleontologists and evolutionary biologists as well as the public.

## Introduction

Cnidarians are generally accepted to be the sister group to bilaterians (*Brusca et al., 2016*; *Erwin et al., 2011*; *Leclère and Röttinger, 2016*; *Zapata et al., 2015*) and are represented by a huge variety of jellyfish, sea anemones, corals, sea fans, hydrozoans (including the colonial siphonophores) and less familiar parasitic groups (*Raikova, 1988*). Although often sessile (polyps) or parasitic, many of them are motile animals and a large proportion of them (such as jellyfish) use muscles to move very actively through the water column. In contrast to bilaterians, cnidarians owe most of their contractile power to epitheliomuscular or myoepithelial cells that make up both epithelial body layers (*Brusca et al., 2016*; *Schmidt-Rhaesa, 2007*). These specialized cells contain interconnected contractile basal extensions (myonemes or myofilaments) that altogether form longitudinal or circular sheets and play a role equivalent to the muscle layers of other animals. Epitheliomuscular cells are connected to nerve cells via chemical synapses (*Westfall et al., 1971*). Cnidarian muscles are characterized by multifunctional capacities and plasticity and perform key functions in locomotion, defense from predators, feeding and digestion at all life-cycle stages (planula, polyp, and medusa stages; see *Leclère and Röttinger, 2016*). In medusae, locomotion is achieved by the rhythmic pulsation of circular sheets of epithelial striated muscles located around the bell margins and lining the subumbrellar surface. These contractions are counteracted by the elastic properties and antagonistic force of the mesoglea and result in expulsion of water from beneath the bell and thus displacement of the medusa via jet propulsion (*Brusca et al., 2016*).

The Precambrian fossil record of cnidarians remain sporadic and controversial, although molecular models often predict a very ancient (e.g. pre-Ediacaran) origin of the group (*Erwin et al., 2011*). *Haootia quadriformis* from the Ediacaran (Fermeuse Formation; ca. 560 Ma; Newfoundland, eastern Canada) roughly resembles modern stalked jellyfish, such as staurozoans, and bears very fine wrinkles interpreted as putative coronal muscles (*Liu et al., 2014a*; *Liu et al., 2015*; see also *Miranda et al., 2015*). Numerous circular forms with a radial pattern have been described in the Ediacaran (e.g. *Cyclomedusa*; Ukraine, Russia; see *Zaika-Novatskiy et al., 1968*; *Fedonkin, 1981*). Although some of them potentially represent bell imprints of jellyfish, others probably have a different origin (e.g. circular holdfasts of non-cnidarian sessile organisms or possible gas-escape sedimentary structures; see *Sun, 1986*). Conulariids is an extinct group of marine animals characterized by a hard exoskeleton resembling a set of morphological features with modern jellyfish such as a tetramerous symmetry, grooved corners and a periderm with numerous transverse ribs. Conulariids have been resolved as stem-group Scyphozoa (*Van Iten et al., 2006*) and have very likely ancestors in the Precambrian, such as *Vendoconularia triradiata* and *Paraconularia* sp. from the terminal Ediacaran of Russia (*Van Iten et al., 2005*; *Ivantsov, 2017*) and Brazil (*Van Iten et al., 2014*), respectively. *Corumbella werneri* also supports a Precambrian origin of cnidarians, its annulated tube with a square cross-section and a lamellar microfabric resembling that of conulariids (*Pacheco et al., 2015*). Convincing evidence for ancestral jellyfish-like medusozoans comes from the early Cambrian Chengjiang Lagerstätte (ca. 521 Ma; Yunnan Province China) and is best exemplified by *Yunnanoascus haikouensis*, which shares a set of morphological features with modern jellyfish such as a tetramerous symmetry, rhopalia, long tentacles around the bell margin, and a possible manubrium in the central part of the bell (*Han et al., 2016a*). Although Chengjiang fossils usually show extremely fine details of soft animal tissues and organs (including digestive, nervous, and reproductive systems), muscles remain elusive, and no trace of possible coronal muscles can be seen in *Yunnanoascus*. Other jellyfish from the mid-Cambrian Marjum Formation (ca. 505 Ma, Utah, USA; see *Cartwright et al., 2007*) display fine recognizable anatomical details such as the exumbrella and subumbrella, tentacles and relatively well-preserved coronal muscles that suggest swimming capacities.

The Kuanchuanpu Formation (ca. 535 Ma, lowermost Cambrian Fortunian Stage; Shaanxi Province, south China) yields a great variety of three-dimensionally preserved microfossils including cnidarians such as Olivooidae (*Dong et al., 2013*; *Dong et al., 2016*; *Han et al., 2013*; *Han et al., 2016b*; *Liu et al., 2014b*; *Liu et al., 2017*; *Steiner et al., 2014*). The developmental sequence of *Olivooides* starts with a spherical embryo that, after hatching, gives rise to a conical juvenile (*Bengtson and Yue, 1997*), suggesting direct development (*Han et al., 2013*; *Steiner et al., 2014*; *Wang et al., 2020*) with no counterpart among modern cnidarians. However, microtomography clearly shows that post-embryonic *Olivooides* does have anatomical features typical of modern cnidarians, a radial symmetry, single body opening, exumbrella and subumbrella, interradial septa (internal ridges), gonads, manubrium, oral

lips, apertural lobes, tentacles, frenula and velaria (*Dong et al., 2013*; *Dong et al., 2016*; *Han et al., 2013*; *Han et al., 2016b*; *Wang et al., 2020*), which collectively support its placement and that of related forms within Medusozoa (*Dong et al., 2013*; *Dong et al., 2016*; *Han et al., 2013*; *Han et al., 2016b*; *Liu et al., 2014b*; *Liu et al., 2017*; *Wang et al., 2020*). *Wang et al., 2017* reported possible coronal muscles around the aperture of *Sinaster* (Olivooidae) but did not investigate their organization and possible function. We describe here secondarily phosphatized muscle fibers preserved in three dimensions, in post-embryonic stages of olivooids from the early Cambrian (Fortunian) Kuanchuanpu Formation. They represent the oldest occurrence of muscle tissue in cnidarians and more generally in animals. We also address the nature (e.g. myoepithelial) and function of this muscle system through detailed comparisons with modern jellyfish.

These new findings prompted us to re-examine and integrate fossil data obtained from other early Cambrian groups such as ecdysozoans (e.g. worms, lobopodians; see *Budd, 1998*; *Vannier and Martin, 2017*; *Young and Vinther, 2017*; *Zhang et al., 2016*), which together shed light on the diversity and functions of muscle systems in early animals.

## Results

The 12 fossil specimens studied here have the diagnostic features of post-embryonic olivooids (*Dong et al., 2013*; *Dong et al., 2016*; *Han et al., 2013*; *Han et al., 2016b*; *Wang et al., 2017*; *Wang et al., 2020*), such as an ovoid shape, pentaradial symmetry and the presence of a periderm, apertural lobes, exumbrella, perradial ridges, and interradial furrows (*Figure 1—figure supplements 1 and 2*; Figure 2A-E; *Figure 2—figure supplement 1*). They also display a well-preserved network of circular fibers (*Figures 1 and 2A–E*), tentatively interpreted as coronal muscles by *Wang et al., 2017* in a pilot study.

The body has a consistent ovoid shape and size (diameter between 560 and 580 μm) and is enclosed by a smooth periderm (5–10 μm in thickness) (*Figure 1—figure supplement Figure 1—figure supplements 1A and 2C, E-H*). Centripetal, triangular projections, termed perradial apertural lobes (see *Han et al., 2013*; *Wang et al., 2020*) can be seen around the oral aperture of most specimens. They correspond to perradii and are organized with pentaradial symmetry (*Figure 1—figure supplements 1 and 2*; *Figure 2—figure supplement 1A, C*). The partial loss of the periderm and perradial apertural lobes in numerous specimens reveals a fine network of underlying closely packed, circular fibers, that are best developed around the oral aperture where they form four or five separate concentric bundles (individual thickness between 9 and 15 μm), each consisting of numerous (possibly up to eight) individual fibers (*Figures 1B,C and 2B,C,E and F*). These fiber rings run around the oral aperture, have a consistent thickness and do not seem to be interrupted (*Figure 1—figure supplement 1*; *Figure 2A, D*, *Figure 2—figure supplement 1A, B*). Comparable fibers occur all over the body, but seem to be sparser toward the aboral pole (*Figure 1—figure supplement 1B, C* and *Figure 1—figure supplement 2A-F*; *Figure 2D*) and not organized in well-defined bundles (*Figure 1D-G*; *Figure 2—figure supplements 1 and 2*; *Figure 2A*). Individual fibers are cylindrical (diameter ca. 2 μm), lying mostly parallel to each other, although oblique V-shaped interconnections (angle ca. 20–30°) may occur locally (*Figure 2C*). Fibers are finely and evenly coated with microcrystalline (ca 0.4 μm) calcium phosphate (*Figure 1B, C and E*). Circular fibers clearly extend into the triangular perradial apertural lobes (*Figure 1—figure supplement 1A, E, F*; *Figure 2A and D*). The circular fiber network seems to be overprinted by faint longitudinal depressions in the aboral half (*Figure 2A, D*, *Figure 2—figure supplement 1B*) that may correspond to interradial furrows.

## Discussion
### Myoepithelial muscles in cnidarians from the basal Cambrian

The close-knit circular fibers found in the bell of Olivooidae sp. can be confidently interpreted as epithelial muscle fibers based on their individual cylindrical shape, size range (around 2 μm in diameter), regular arrangement in bundles (e.g. possibly five more around the bell margin), and closely-packed distribution over a single anatomical surface corresponding to the subumbrella (*Figure 1*; *Figure 2—figure supplement 2*). Our interpretation is also strongly supported by close similarities with the muscular system of modern cnidarians (myoepithelial cells; MEC). For example, the medusae of hydrozoans (*Figure 2G–I*; *Figure 3A*) display a continuous network of circular striated and radial

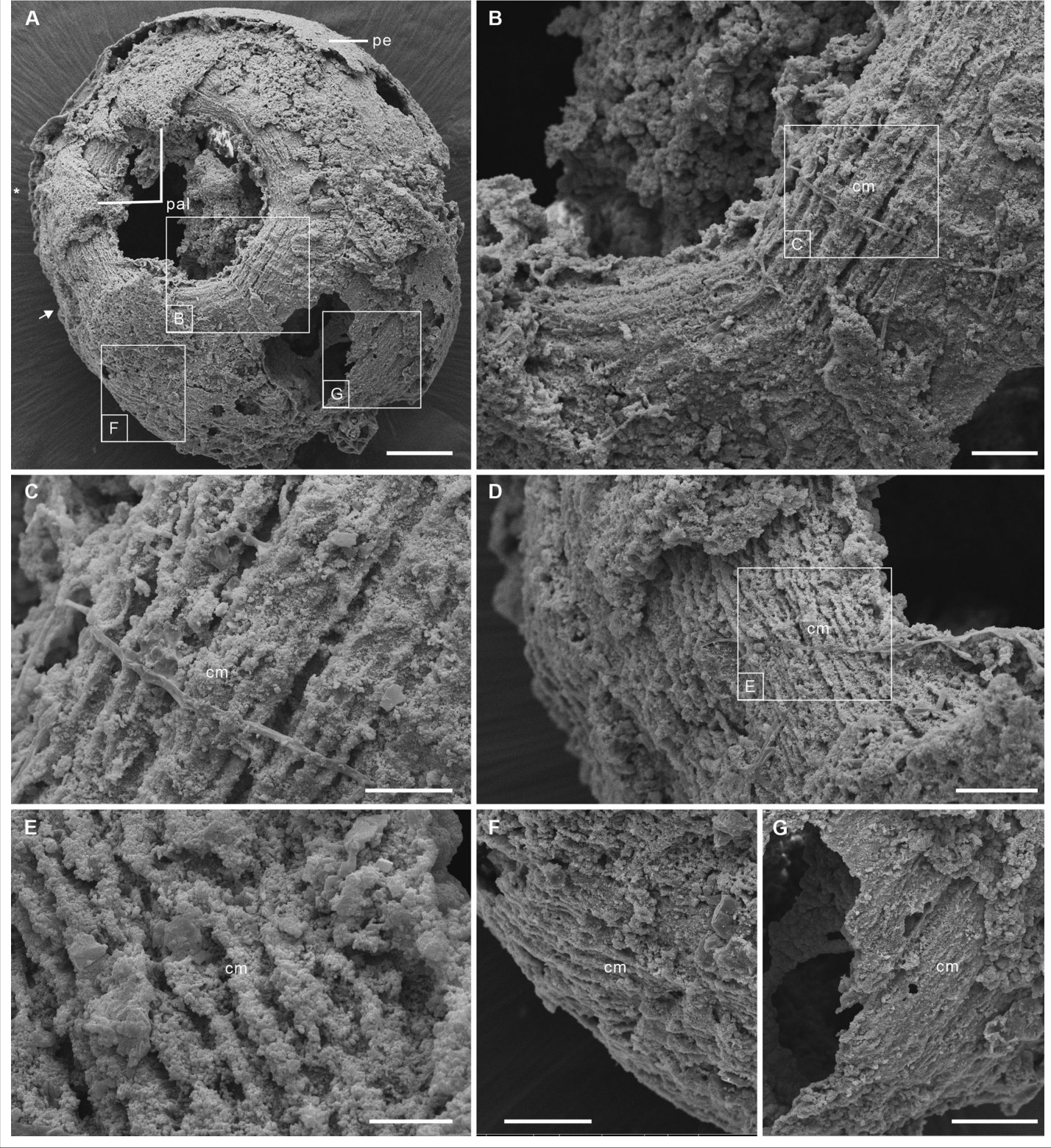

**Figure 1.** Post-embryonic stage of *Olivooides* sp. from the early Cambrian Kuanchuanpu Formation (ca. 535 Ma; South China), showing exposed muscle fibers. ELISN150-278. Scanning electron micrographs.

(**A**) General view of oral side. (**B**) Details of fiber bundles around aperture (location indicated in **A**). (**C**) Close-up showing individual fibers within each bundle. (**D**) Dense network of fibers (location indicated in **A**). (**E**) Close-up of individual fibers coated with fine grains of calcium phosphate. (**F**), (**G**) Circular fibers approximately half way between the oral and aboral poles. Abbreviations: cm, circular muscle; pal, perradial apertural lobe; pe,

*Figure 1 continued on next page*

*Figure 1 continued*

periderm; \*, perradii; →, interradii. Scale bars: 100 µm in (**A**); 20 µm in (**B**); 10 µm in (**C**), (**E**); 20 µm in (**D**), (**F**) and (**G**).

The online version of this article includes the following figure supplement(s) for figure 1:

**Figure supplement 1.** Three scanning electron micrographs of the *Olivooides* specimens.

**Figure supplement 2.** Four scanning electron micrographs of the *Olivooides* specimens.

**Figure supplement 3.** Polyp of Coronatae sp. from South China Sea showing the periderm.

**Figure supplement 4.** Origin of the fossil material.

smooth muscles covering the underside of their bell (subumbrella). These 1-to-4-µm-thick individual fibers show oblique interconnections (*Figure 2G-I*; *Figure 2—figure supplement 2*), and sparser radial smooth fibers run perpendicular to them (*Leclère and Röttinger, 2016*). A very similar configuration can be seen in early Cambrian olivooids, which display continuous circular fibers (in some cases covered by transversal structures imprinted on the circular bundles; see *Figures 1–3A–C*). Microscopic series of functional units (sarcomeres) that characterize striated muscles and give them a typical striated appearance (*Figure 2G-I*; *Figure 2—figure supplement 2*) are not discernible in the muscle network of Cambrian olivooids, making it impossible to class them as striated or smooth (*Schmidt-Rhaesa, 2007*). The muscles of olivooids seem to be more developed and concentrated around the bell margin, as seen in modern jellyfish, such as *Clytia*, *Pelagia*, and *Chrysaora* (*Figure 2G-I*; *Figure 2—figure supplement 2*). Other olivooids from the Kuanchuanpu Formation such as *Sinaster* have a comparable concentration of strong muscles around the oral aperture (possibly five bundles; see *Wang et al., 2017*) but, unlike those of the present specimens, seem to be interrupted by interradial structures. The muscle fibers of olivooids distribute over a surface interpreted as the inner layer of the bell (subumbrella), as in modern medusae. In contrast with the distinct muscle system of bilaterians, modern cnidarians have myoepithelial cells that are fully integrated into the ectodermal and endodermal epithelial tissues. Although the cellular organization cannot be seen in the present fossil material, we hypothesize that the muscular system of Cambrian olivooids was similarly composed of myoepithelial cells that had myofilaments projecting from their basal side. The circular network of olivooids may have been supplemented by longitudinal muscles accommodated within adradial furrows (*Figure 3C*). However, no clear individual fibers could be distinguished in these adradial areas possibly owing to decay or taphonomy. Paired features interpreted as tentacular buds occur around the oral rim of some olivooids (*Han et al., 2013*; *Wang et al., 2020*). Their external annulations may represent underlying muscle fibers (*Wang et al., 2020*), or, more likely, anchoring features of nematocysts. The tentacles of modern cnidarians have longitudinal muscles but lack circular fibers (*Hyman, 1940*).

The current, well-accepted, hypothesis is that olivooids developed from an ovoid post-embryonic form (present material) into a conical corrugated polyp-like structure (*Bengtson and Yue, 1997*; *Dong et al., 2016*; *Han et al., 2013*; *Liu et al., 2014a*; *Steiner et al., 2014*; *Wang et al., 2020*). The transition to polyps is characterized by the gradual increase of external ornamented ridges that most probably resulted in a complete anatomical reorganization, as commonly seen in the life cycle of modern cnidarians. Unfortunately, very little is known of the internal anatomy of these polyps, except that they secreted a tubular feature (periderm) comparable with that of some extant scyphozoans (*Figure 1—figure supplement 3*; *Song et al., 2021*) and had possible oral lobes (*Wang et al., 2020*). Although rare, clear traces of circular fibers do occur in the polyps of *Olivooides mirabilis* (see Figure 12 in *Steiner et al., 2014*), suggesting that features of the muscle system were conserved through the lifecycle.

In summary, the close-knit fibrous network described here in post-embryonic olivooids is the oldest record of a muscular system in cnidarians and more generally in animals.

## Functions of muscles in post-embryonic olivooids

The occurrence of strong muscles around the bell aperture and inside the perradial apertural lappets suggests that olivooids could contract their bell as modern medusae do (*Figures 3A–C , and 4A–C*). In extant jellyfish, these contractions are counteracted by the elastic properties and antagonistic force of the adjacent mesoglea. As a result, water is rhythmically expelled from beneath the bell and drives the medusa through the water column via jet propulsion (*Brusca et al., 2016*). Although olivooids

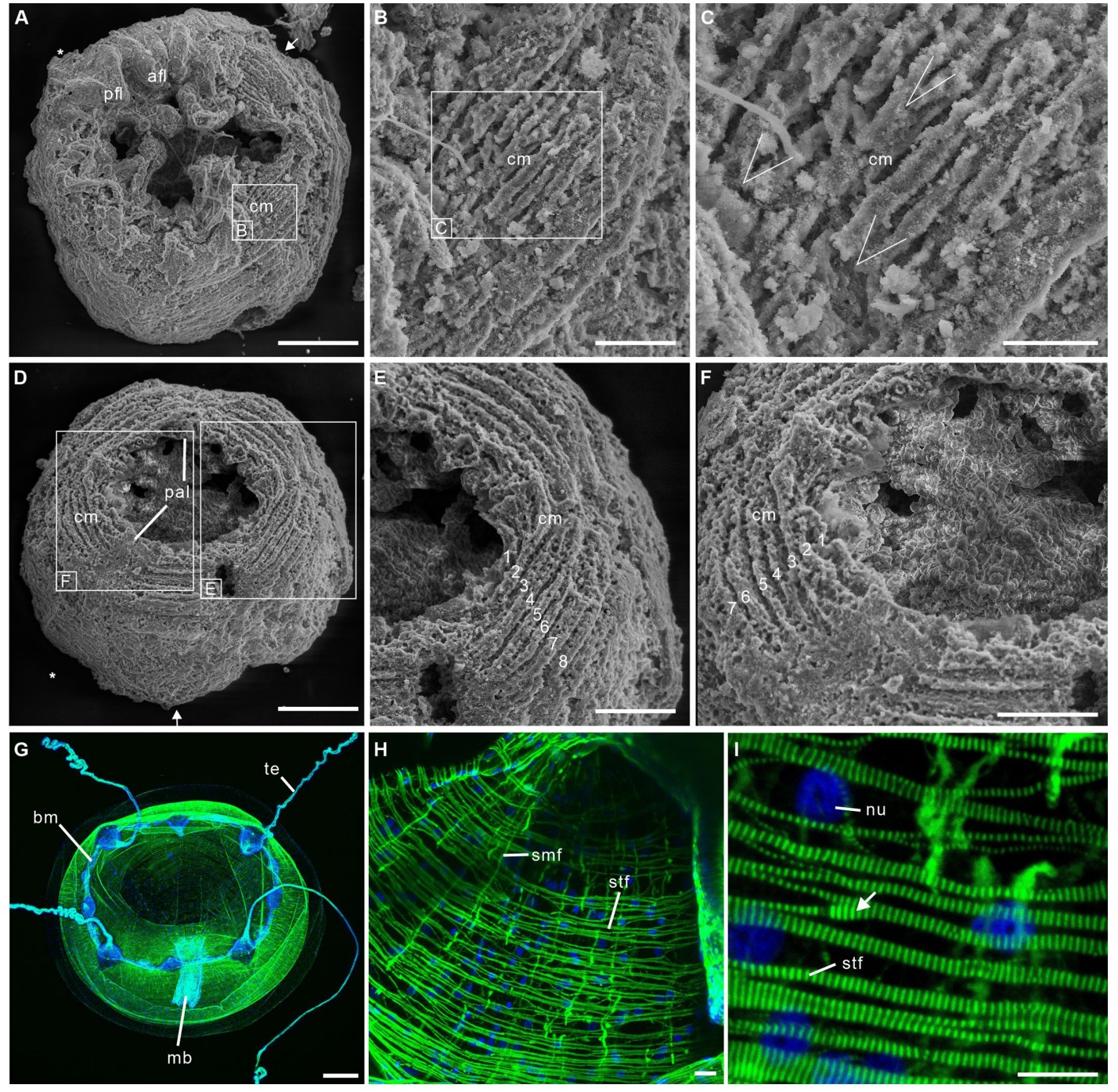

**Figure 2.** Post-embryonic stage of *Olivooides* sp. from the early Cambrian Kuanchuanpu Formation (ca. 535 Ma; South China), showing exposed muscle fibers by SEM. (A–F) and myoepithelial muscle network in extant hydrozoan jellyfish by fluorescence microscopy (G–I). (A–C) ELISN052-33. General view of oral side and details of apertural circular muscle fibers and the V-shaped interconnection between the fibers in (C). (D–F) ELISN061-19. General view of oral side and details of apertural, circular muscles fibers. (G–I) *Eirene sp.* (Hydrozoa) young medusa, general oral view, circular muscles along subumbrella and details of striated fibers; white arrow (I) indicates bifurcating fibers. Green and blue in (G–I) correspond to actin (phalloidin) and DNA (Hoechst) staining. Abbreviations: afl, adradial fold lappet; bm, bell margin; cm, circular muscle; mb, manubrium; nu, nucleus; pal, perradial apertural lobe; pfl, perradial fold lappet; smf, smooth (radial) muscle fiber; stf, striated (circular) muscle fiber; te, tentacle *, perradii; →, interradii. Scale bars: 200 μm in (A), (D); 100 μm in (G); 50 μm in (E), (F); 20 μm in (B); 10 μm in (C), (H) and (I).

The online version of this article includes the following figure supplement(s) for figure 2:

**Figure supplement 1.** Two scanning electron micrographs of the *Olivooides* specimens.

**Figure supplement 2.** Myoepithelial muscle network in extant medusozoan cnidarians.

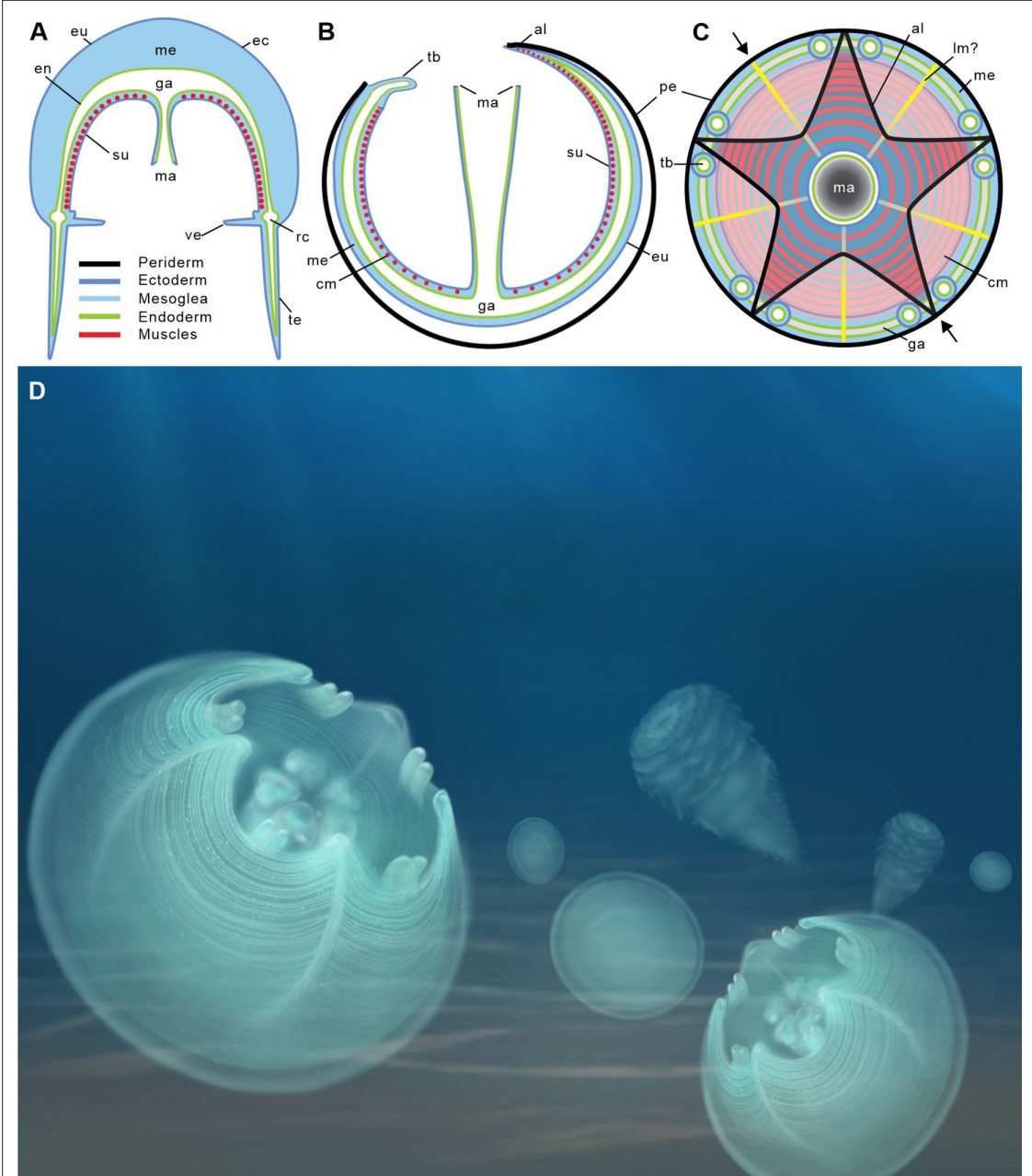

**Figure 3.** Location of epithelial muscles in extant hydromedusae (**A**) and early Cambrian Olivooidae medusozoans (**B, C**). (**A**), (**B**) Simplified radial sections through body. (**C**) In oral view. (**D**) Artistic reconstruction of 535-million-year-old olivooid cnidarians showing eggs (the prehatched, background), post-embryonic (foreground) and polyp (background) stages. The circular muscle system is visible through the translucent periderm. The location of section (**B**) is indicated in (**C**) by small black arrows. Abbreviations: al, apertural lobe; cm, circular (coronal) muscle; en, endoderm; eu, exumbrella; ga, gastrovascular cavity; lm?, possible longitudinal muscle; ma, manubrium; me, mesoglea; pe, periderm; rc, radial canal; su, subumbrella; tb, tentacular bud; te, tentacle; ve, velum. Not to scale.

share important external and internal morphological features with medusozoans (*Dong et al., 2013*; *Han et al., 2013*; *Han et al., 2016b*; *Wang et al., 2017*; *Wang et al., 2020*), they are distinguished by an unusual pentaradial symmetry and life cycle (with an ovoid post-embryonic stage developing into a conical polyp). This life cycle has no direct counterpart in modern medusozoans (e.g. *Brusca et al., 2016*) that develop from a motile planula larva into a polyp and eventually a juvenile medusa through various processes (e.g. strobilation generating ephyrae in scyphozoans; see *Gershwin, 1999*). It has been suggested (*Wang et al., 2020*) that the post-embryonic stage of olivooids combined the

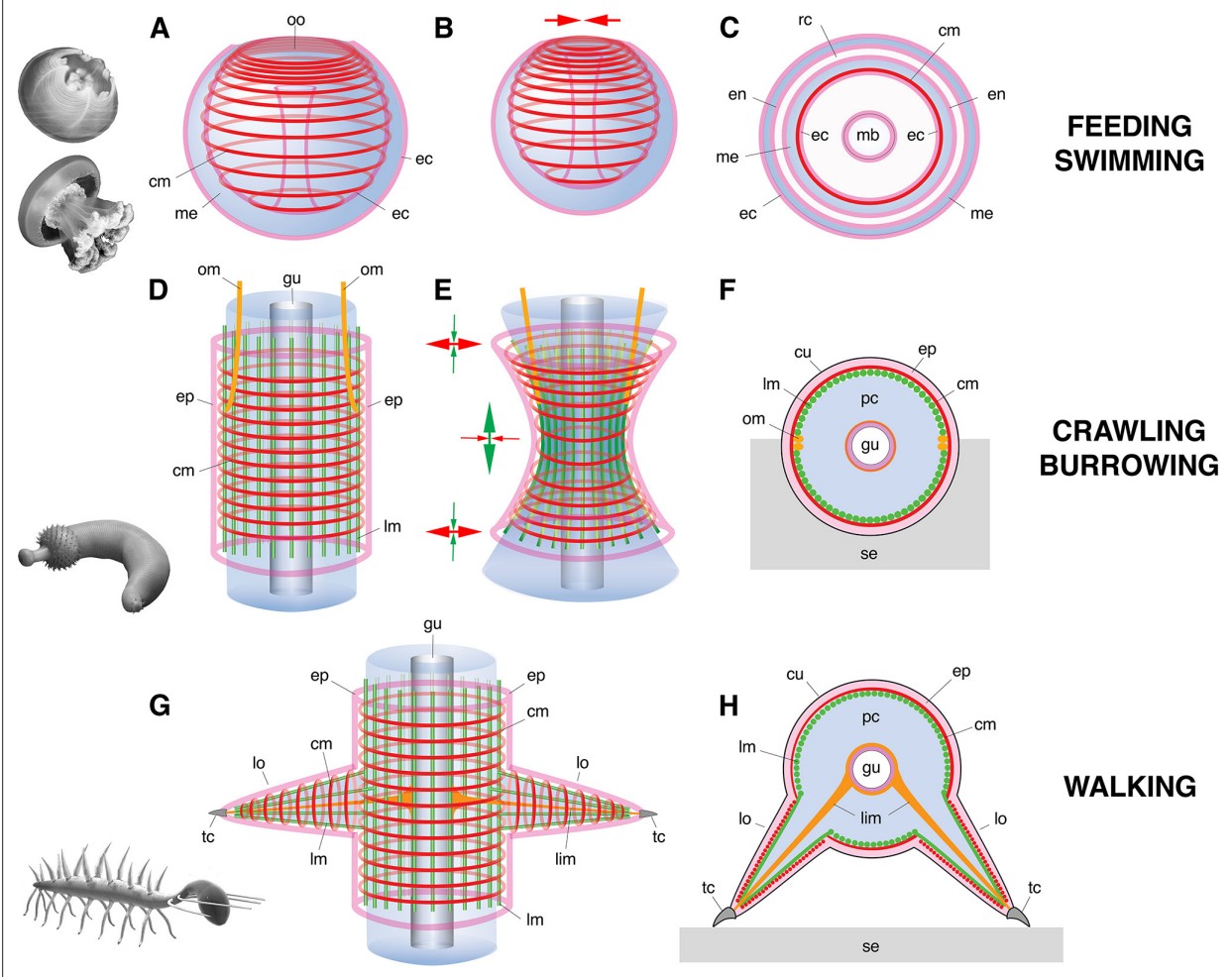

**Figure 4.** Basic muscle systems in early Cambrian animals and their main functions. (**A–C**) Contractile epithelial muscles (MEC, myoepithelial, predominantly circular) and antagonistic mesoglea exemplified by olivooiids and free-swimming jellyfish; idealized relaxed (**A**) and contracted (**B**) states and simplified transverse section (**C**). (**D–F**) Grid-like network of circular and longitudinal subepidermal muscle fibers (HMS) around a cylindrical body filled with antagonistic internal fluid (hydroskeleton) exemplified by scalidophoran worms: idealized relaxed state (**D**); peristaltic contractions along body (**E**) and transverse section (**F**). (**G–H**) Longitudinal, circular muscles and extrinsic retractor muscles in lobopodians: idealized relaxed state (**G**); transverse section (**H**). Images (from top to bottom) represent an olivooid cnidarian, an extant jellyfish, the scalidophoran worm *Ottoia* (see *Vannier, 2012*) and the lobopodian *Hallucigenia* (see *Chen and Zhou, 1997*). Drawings and images not to scale. Abbreviations: cm, circular muscle; cu, cuticle; ec, ectoderm; en, endoderm; ep, epidermal layer; gu, gut; lc, lobopod claw; lim, limb muscle; lm, longitudinal muscle; lo, lobopod (soft leg); mb, manubrium; me, mesoglea; om, oblique muscle; oo, oral opening (bell margin); pc, primary cavity filled with fluid; rc, radial canal; se, sediment; tc, terminal claw.

The online version of this article includes the following figure supplement(s) for figure 4:

**Figure supplement 1.** Muscle system in priapulid worms (Ecdysozoa).

**Figure supplement 2.** Muscle system in Cambrian lobopodians (Ecdysozoa, Panarthropoda).

characters of a medusa with those of a polyp and thus would resemble a juvenile sessile jellyfish almost encased within a periderm, with its bell aperture facing upwards. At first sight, these circular muscles may suggest a role in locomotion, as in modern medusae. However, major structural differences separate modern jellyfish from Cambrian olivooids (see above). Whereas the movement of modern medusae is unconstrained, that of olivooids was most probably strongly hindered by its periderm (*Bengtson and Yue, 1997*; *Steiner et al., 2014*). The assumed mesoglea layer of olivooids seems to have been very thin (narrow gap between ex- and subumbrella; see *Han et al., 2013*; *Han et al., 2016b*; *Dong et al., 2016*; *Wang et al., 2017*; *Wang et al., 2020*) thus limiting its dynamic capacity. Swimming efficiency of modern medusae depends on the power and distribution of muscles but also largely on the flexibility of the bell margin, a condition that is not found in olivooids. Powerful muscle

contractions may have been able to propel the animal over a very short distance but are unlikely to have sustained dynamic locomotion through the water column. This hypothetical 'clumsy' locomotion would have probably created more disadvantages (e.g. energy cost) than advantages to the animal. Moreover, it is unlikely to have generated any adequate escape reaction (e.g. from predators) or effective feeding technique. We favor an alternative option. One of the most vital requirements for post-embryonic olivooids was probably to obtain sufficient food. Modern medusae feed on small soft-bodied prey by using nematocyst-laden tentacles and oral arms that convey food to the mouth (*Brusca et al., 2016*). Post-embryonic olivooids had tentacular buds that were not enough developed to capture food. Feeding may have been achieved by a different method, such as the rhythmic contractions of their circular muscles, especially those bordering the bell aperture (*Figure 4A–C*). Such contractions may have resulted in pumping and engulfing sea water containing potential food particles. Extant jellyfish such as *Aurelia* feed by a comparable mechanism (*Costello and Colin, 1994*). The aperture lappets of olivooids may have played a key role in closing the aperture after contraction in retaining food particles within the bell cavity before being transferred to the mouth and preventing foreign matters from entering the bell. Besides, the remarkable ability of modern cnidarians to transform and rearrange their muscle systems (e.g. transition from polyp to medusa stages; *Brusca et al., 2016*) may have played in this evolution and diversification. Muscle fibers primarily assigned to feeding (olivooids) may have been used for other functions such as active propulsion and escape responses during evolution while maintaining an important role in prey capture. In this context, we could make the bold speculation that the origin of medusae swimming, associated with the subsequent loss of periderm, is an exaptation of rhythmic feeding and respiration among benthic periderm-dwelling medusozoans, probably during Ediacarian-Cambrian period (*Figure 4A–C*).

## Early evolution of muscle systems

The shift of the actin-myosin system from strictly intracellular to intercellular functions seems to have occurred in the early stages of metazoan evolution (*Schmidt-Rhaesa, 2007*). Although the most basal metazoans lack true muscles they do have the capacity to contract their body or part of it. For example, sponges have myocytes around the osculum that play a role in expelling wastes (*Bagby, 1966*). The fiber cells of placozoans such as *Trichoplax adherens* have contractile extensions packed with actin filaments that link the ventral and dorsal epithelia (*Armon et al., 2018*). Although the gliding motility of placozoans is mainly performed by ciliated epithelial cells, fiber cells seem to be involved in active body deformation and invagination possibly related to feeding (*Smith and Mayorova, 2019*). The integration of the actin-myosin system within a dense network of myoepithelial cells as seen in Cambrian and modern cnidarians is likely to have been a major innovation that provided animals with new capacities for powerful movements (e.g. swimming, feeding, respiration, etc.). We suggest that the myoepithelial fiber network seen in post-embryonic olivooids may represent one of the oldest and basic types of animal muscle systems that probably has its origin in the late Precambrian. Larger jellyfish from the early (e.g. *Han et al., 2016a*) and mid-Cambrian (Marjum Lagerstätte; coronal muscles, *Cartwright et al., 2007*) are likely to have co-opted this basic system for more diverse functions such as swimming within the water column (e.g. active jet propulsion) and prey capture as seen in modern jellyfish such as *Aurelia*.

## Diversity of muscle systems and locomotion modes in early Cambrian animals

Ecdysozoans (worms and panarthropods) provide additional evidence on the diversity of muscle systems in early Cambrian animals.

Scalidophoran worms were diverse and numerically abundant in the Cambrian (e.g. Burgess-shale-type Lagerstätten). Modern representatives of the group such as *Priapulus* have a close-knit network of circular and longitudinal muscle lining the inner surface of the body wall, that is surrounded by ECM (*Figure 4D, F*; *Figure 4—figure supplement 1*) and often termed 'Hautmuskelschlauch' (HMS; see *Schmidt-Rhaesa, 2007*). The enclosed mass of incompressible fluid (hydroskeleton) that fills the primary cavity of such worms is the principal antagonist for muscular action (*Figure 4*). Dynamic interactions between HMS and hydroskeleton allow them to perform repeated body contraction and extension for burrowing (*Figure 4D–F*) (e.g. *Vannier et al., 2010*) and feeding. The eversion of the introvert is a process of pumping body fluids into the anterior

body region, whereas its invagination is performed by extrinsic oblique retractor muscles. Based on remarkable anatomical and functional similarities with modern priapulids (*Figure 4D, F*; *Figure 4— figure supplement 1*), we posit that Cambrian scalidophorans also had a HMS-type musculature and possible retractors. This hypothesis is supported by abundant crawling and burrowing traces found in the basal Cambrian (e.g. treptichnid burrow systems; see *Kesidis et al., 2019*; *Vannier et al., 2010*) and the late Precambrian (e.g. *Evans et al., 2020*; *Gehling et al., 2001*) that could not have been made without the action of HMS on a hydrostatic skeleton (*Figure 4D, F*; *Figure 4— figure supplement 1*). HMS clearly differs from the musculature of olivooid cnidarians in at least two key features: (1) HMS does not consist of myoepithelial cells and (2) the antagonist onto which muscular force transferred is not the mesoglea but the primary cavity filled with fluid (*Leclère and Röttinger, 2016*).

Lobopodians (e.g. *Ortega-Hernández, 2015*) is an informal group of ecdysozoans with an annulated cuticle and paired soft legs (lobopods), that is crucial for understanding the remote ancestry of euarthropods. They are best exemplified by iconic Cambrian forms such as *Hallucigenia* (*Smith and Caron, 2015*; *Smith and Ortega-Hernández, 2014*) and *Microdictyon* (*Pan et al., 2017*). Most recent phylogenetic trees (e.g. *Aria et al., 2021*; *Giribet and Edgecombe, 2019*) have resolved Cambrian lobopodians as an 'intermediate' group between scalidophoran worms and arthropods with an arthropodized exoskeleton. The inner surface of their body wall was lined with closely packed circular and longitudinal muscle fibers that seem to have extended into the limbs (see *Tritonychus* in *Zhang et al., 2016*; *Figure 4G, H*; *Figure 4—figure supplement 2A, B*). This configuration strongly recalls that of ecdysozoan worms (see above). In *Paucipodia* from the Chengjiang Lagerstätte (see *Vannier and Martin, 2017*; see *Figure 4—figure supplement 2C-E*), a connecting strand runs between the terminal claw of the limbs and the area surrounding the gut and is interpreted here as a possible retractor muscle (see analogues in extant onychophorans *De Sena Oliveira and Mayer, 2013*). Although unsegmented as in smaller lobopodians, larger Cambrian lobopodians such as *Pambdelurion* from the Sirius Passet Lagerstätte (ca. 520 Ma) are characterized by a more complex musculature with paired, lateral, ventral and dorsal longitudinal muscles. Well-developed bundles of extrinsic and intrinsic limb muscles presumably controlled their leg motion as in modern onychophorans (*Budd, 1998*; *Hoyle and Williams, 1980*; *Young and Vinther, 2017*; see *Figure 4G and H*).

Early euarthropods that co-existed with lobopodians and scalidophoran worms (e.g. Burgess-Shale-type Lagerstätten) had already acquired rigid exoskeletal elements (sclerotized cuticular elements jointed by an arthrodial membrane, such as body sclerites and appendage podomeres) that were operated by a lever-like system of segmentally arranged antagonistic muscles as seen in *Kiisortoqia* and *Camparamuta* from the Sirius Passet Lagerstätte (*Young and Vinther, 2017*). This suggests that the rise of euarthropods was associated with a profound rearrangement of the muscle system possibly inherited from lobopodian ancestors, such as the reduction of the circular HMS musculature that lost its primary hydrostatic function and peristaltic capabilities because of exoskeletal rigidity (*Young and Vinther, 2017*).

Deuterostomes also evolved muscle systems during the Cambrian. Well-developed sigmoidal blocks occur within the trunk of early and mid-Cambrian chordates such *as Myllokunmingia* (*Shu et al., 1996*) and *Haikouichthys* (*Shu et al., 2003*; possibly craniate) *and Pikaia* (*Conway Morris and Caron, 2012*; stem-group chordate). They recall the W-shaped wall musculature of modern cephalochordates such as *Branchiostoma* and fish, which is derived from segmental coelomic compartments. Although boundaries between adjacent blocks (myosepta) are often well defined, none of these 'fishlike' fossils reveals details on the nature and arrangement of muscle fibers. These assumed muscle blocks occupied much of the body wall of these ancient chordates and most probably played a crucial role in swimming.

In summary, both fossil and indirect evidence presented here indicate that different types of musculature co-existed among early Cambrian animals: (1) myoepithelial circular (MEC) muscles in cnidarians, (2) grid-like and subepidermal (HMS) muscles in scalidophoran worms, (3) HMS and extrinsic muscles to control leg motion in lobopodians, (4) segmentally arranged muscles tightly integrated to exoskeletal elements in early euarthropods, (5) well-developed W-shaped muscles in early chordates.

Whereas the myoepithelial system appears as the most basic one, that of ecdysozoans seems to have undergone considerable changes and diversification over a relatively short time interval during the early Cambrian. This remarkable diversity and plasticity of muscle systems allowed a great variety

of animals to explore and colonize new environments and can be seen as one of the driving forces of the animal radiation.

## Materials and methods

### Geological setting

All studied fossils come from phosphatic limestones collected from the Kuanchuanpu Formation at the Shizhonggou section (Ningqiang County, Shaanxi Province, China; see *Figure 1—figure supplement 4*). Biostratigraphy (*Anabarites – Protohertzina – Arthrochites* zone; *Qian, 1977*; *Qian, 1999*) indicates that these rocks correspond to the Meishucunian Stage that is the equivalent of the lowermost Cambrian Terreneuvian Stage. Radiochronology (U-Pb method; *Sawaki et al., 2008*; *Peng et al., 2012*) confirms that the Kuanchuanpu Formation is approximately 535 Ma. Secondarily phosphatized fossils were extracted from rocks via a standard acid digestion in 7% acetic acid. Dried residues with a grain-size >60 μm were sorted and picked under a binocular microscope. Twelve specimens of Olivooidae (Cnidaria) bearing well-preserved muscle fibers were selected for the present study and mounted for SEM (FEI Quanta 400 FEG scanning electron microscope at Northwest University, China; Au-coating, high-vacuum). They belong to *Sinaster petalon Wang et al., 2017* (ELISN115-39), *Hanagyroia orientalis Wang et al., 2020* (ELISN107-470) and Olivooidae sp. (ELISN150-278, ELISN111-54, ELISN052-33, ELISN045-143, ELISN012-16, ELISN061-19, ELISN087-64, ELISN088-48, ELISN087-33 and ELISN098-19). All specimens are deposited in the collections of the Shaanxi Key Laboratory of Early Life & Environments and the Department of Geology, Northwest University, China ('ELI' is an acronym of the former Early Life Institute that is now replaced by 'ELISN', SN for Shaanxi Province, China).

Two-week-old *Clytia hemisphaerica* medusae, newly released *Eirene sp.* medusae, 1-month-old *Chrysaora colorata* and *Pelagia noctiluca* metaephyrae were raised in the laboratory (Villefranche-sur-mer, France) following *Lechable et al., 2020* and *Ramondenc et al., 2017* culture protocols. Fixation followed by Phalloidin (actin) and Hoechst (nuclei) staining was performed on the four species as described for *Clytia hemisphaerica* in *Sinigaglia et al., 2020*. Samples were mounted in 50% Citifluor AF1 antifade mountant and imaged using Leica SP8 confocal and Zeiss Axio-Observer microscopes.

Extant priapulid worms (*Priapulus caudatus*) (see *Figure 4—figure supplement 1*) were collected (JV) from near the Kristineberg Marine Station (Sweden), fixed with glutaraldehyde and dried (Critical Point) for SEM observations (Univ. Lyon).

## Acknowledgements

We thank the China Postdoctoral Science Foundation (No. 2020M672013), the Natural Science Foundation of China (Nos. 41902012, 41720104002, 41876180, 41772021, 41911530236), the Strategic Priority Research Program of the Chinese Academy of Sciences (No. XDB26000000), 111 Project of the Ministry of Education of China (Nos. D17013, D163107), the Most Special Fund from the State Key Laboratory of Continental Dynamics, Northwest University, China (BJ11060). This work was also supported by Agence Nationale de la Recherche (ANR): Lucas Leclère ANR-19-CE13-0003, the Région Auvergne-Rhône-Alpes and the Univ. of Lyon (PAI grant to JV), for financial support. We thank J Sun and J Luo for the preparation of microfossils and SEM technical assistance, and the CTμ (University of Lyon) for access to electron microscopy. We thank X-G. Zhang for kindly providing images of Cambrian lobopodians. We thank Kentaro Uesugi analyzing the specimens using the computed x-ray microtomography (XTM) at Tohoku University, Japan, and the synchrotron of Spring-8 in Hyogo, Japan. We thank Evelyn Houliston (LBDV, Villefranche-sur-mer) for insightful remarks and corrections, and the three referees: Min Zhu (Institute of Vertebrate Palaeontology & Palaeoanthropology, Chinese Academy of Sciences), Heyo Van Iten (Department of Geology, Hanover College, Indiana) and Bruce Lieberman (Department of Ecology & Evolutionary Biology, University of Kansas). We thank George Perry as the Senior Editor, Tara Bristow, Emma Darkin, Sam Porteous and Nicola Adamson as the Editorial Assistant for correcting and improving our manuscript. We thank Alexandre Jan for raising the medusae and the Paris Aquarium for providing the Eirene and Chrysaora polyp stains. We thank the Marine Resources Centre (CRBM and PIV imaging platform) of Institut de la Mer de Villefranche (IMEV), supported by EMBRC-France. The CRBM is supported by EMBRC-France, whose French

state funds are managed by the ANR within the Investments of the Future program under reference ANR-10-INBS-02.

## Additional information

### Funding

| Funder | Grant reference number | Author |
|---|---|---|
| China Postdoctoral Science Foundation | No. 2020M672013 | Xing Wang |
| National Natural Science Foundation of China | Nos. 41902012 | Jian Han |
| the Strategic Priority Research Program of the Chinese Academy of Sciences | No. XDB26000000 | Jian Han |
| 111 Project of the Ministry of Education of China | Nos. D17013 | Jian Han |
| the Most Special Fund from the State Key Laboratory of Continental Dynamics, Northwest University, China | BJ11060 | Jian Han |
| the Region Auvergne-Rhone-Alpes and the Univ. of Lyon | PAI grant to JV | Jean Vannier |
| Agence Nationale de la Recherche | Lucas Leclère ANR-19-CE13-0003 | Lucas Leclère |
| National Natural Science Foundation of China | 41876180 | Xikun Song |
| National Natural Science Foundation of China | 41720104002 | Jian Han |
| 111 Project of the Ministry of Education of China | D163107 | Jian Han |
| Natural Science Foundation of China | 41772021 | Jian Han |
| Natural Science Foundation of China | 41911530236 | Jian Han |

The funders had no role in study design, data collection and interpretation, or the decision to submit the work for publication.

### Author contributions

Xing Wang, Conceptualization, Data curation, Formal analysis, Investigation, Methodology, Resources, Software, Supervision, Validation, Visualization, Writing - original draft, Writing – review and editing; Jean Vannier, Data curation, Formal analysis, Investigation, Methodology, Software, Supervision, Writing – review and editing; Xiaoguang Yang, Data curation, Formal analysis, Investigation, Methodology, Writing - original draft, Writing – review and editing; Lucas Leclère, Formal analysis, Methodology, Visualization, Writing – review and editing; Qiang Ou, Formal analysis, Methodology, Writing – review and editing; Xikun Song, Formal analysis, Writing – review and editing; Tsuyoshi Komiya, Methodology; Jian Han, Conceptualization, Data curation, Formal analysis, Funding acquisition, Investigation, Methodology, Project administration, Resources, Supervision, Writing – review and editing

### Author ORCIDs

Xing Wang http://orcid.org/0000-0002-1777-864X
Jean Vannier http://orcid.org/0000-0003-0998-1231

Lucas Leclère 
Xikun Song 
Tsuyoshi Komiya 
Jian Han 

## Decision letter and Author response

Decision letter https://doi.org/10.7554/eLife.74716.sa1
Author response https://doi.org/10.7554/eLife.74716.sa2

---

## Additional files

### Supplementary files
• Transparent reporting form

### Data availability

Data for this study are available in the Dryad Digital Repository: https://doi.org/10.5061/dryad.pvmcvdnn1.

The following dataset was generated:

| Author(s) | Year | Dataset title | Dataset URL | Database and Identifier |
|---|---|---|---|---|
| Wang X, Vannier J, Yang X, Leclère L, Ou Q, Song X, Komiya T, Han J | 2022 | Muscle systems and motility of early animals highlighted by cnidarians from the basal Cambrian | https://doi.org/10.5061/dryad.pvmcvdnn1 | Dryad Digital Repository, 10.5061/dryad.pvmcvdnn1 |

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
