## [Editor Report]

Based on exceptionally preserved fossils of olivooids, a group of early cnidarians, from the basal Cambrian of China (535 million years ago), Wang and colleagues reveal primitive muscles of early animals with well-developed system of circular fibers directly comparable with the myoepithelial muscles of modern medusae, representing the oldest record of a muscle system in cnidarians and more generally in animals. The manuscript will be of broad interest to scientists, including paleontologists and evolutionary biologists as well as the public.

---

## [Decision Letter]

**Decision letter after peer review:**

Thank you for submitting your article "Muscle systems and motility of early animals highlighted by cnidarians from the basal Cambrian" for consideration by *eLife*. Your article has been reviewed by 3 peer reviewers, including Min Zhu as Reviewing Editor and Reviewer #1, and the evaluation has been overseen by George Perry as the Senior Editor. The following individuals involved in review of your submission have agreed to reveal their identity: Heyo Van Iten (Reviewer #2); Bruce Lieberman (Reviewer #3).

Essential revisions:

*Reviewer #1:*

Wang and colleagues reported the muscle details in exceptionally preserved fossils of olivooids, a group of early cnidarians, from the basal Cambrian of China (535 million years ago). By comparison to the extant jellyfish, the authors provide an interpretation on functions of these olivooid muscles, and demonstrate the variety of musculature in several early Cambrian invertebrate groups. The inference and the figure presentation in the manuscript are overall clear, and the conclusions are well supported by paleontological data.

*Reviewer #2:*

Using scanning electron microscopy, the authors of this paper have documented some of the earliest evidence of relic muscle tissue in the fossil record. Their interpretations of the histology of their fossil material is based on detailed comparisons of exceptionally preserved relic soft parts in Olivooides, an extremely important early Cambrian genus, with putatively homologous tissues in extant organisms. Their work seems thorough and well-executed, and their interpretations, both of the documented fossil material and of the broader implications of their findings, seem reasonable and well supported by the available evidence. No doubt this study will stimulate further discussion and analysis of the fundamental empirical and theoretical problems in play here, including the timing and consequences of key anatomical innovations in the animal kingdom.

A fascinating study, beautifully illustrated and cogently argued. My only suggestion for revision/improvement is the following: Near the beginning of the introductory paragraph starting with line 60, cite the study of Van Iten et al., (2016) on key cnidarian and putative cnidarian macrofossils (including Haaotia) from the Ediacaran of North and South America (you know the paper). Of particular relevance is Paraconularia sp. From the terminal Ediacaran Tamengo Formation. This fossil constitutes by the far the strongest fossil evidence of the presence of medusozoans (conulariids) during Ediacaran times. So, while the origin of the fine wrinkles in Haootia may be uncertain (and I would go a bit easy on this hypothesis, especially as you seem to be making a similar case for a putative early Cambrian “polyp” or sessile medusa), there is little doubt that medusozoans are present in the Ediacaran fossil record. Moreover, and by analogy with extant coronate polyps (which you illustrate), conulariids presumably had well developed longitudinal and circular muscles.

*Reviewer #3:*

The strengths of the paper are that it very nicely reviews the literature on the early fossil record of medusozoans, it gives a good summary and discussion of previous finds, it provides comprehensive and high quality pictures of the fossils with well abelled illustrations and interpretations, and addresses very important topics in paleontology, evolutionary biology, and invertebrate zoology.

My suggested revisions follow. These involve requests for providing more details on a few matters that could perhaps be viewed as weaknesses, or at least I would judge that making the revisions I have suggested would strengthen aspects of what is already a very strong paper.

1) It would be worth highlighting a bit more how some of the character information that conflicts with the view that olivooids are cnidarians. For instance, their pentaradial symmetry is very problematic. I think it is reasonable to interpret the structures the authors have identified as circular muscles but I think a bit more information as to how these fossils cannot possibly be echinoderms would be very valuable. Would muscles similar to the ones you found not be present in Cambrian (or modern) echinoderms? It would be worth stating or illustrating differences between the two in the revisions. That would provide some evidence supporting what you are arguing is the independent origins of pentaradial symmetry in two different groups.

2) Also, the authors seem to be stating that no radial smooth muscles are found in their fossils. A more definitive statement of this would be valuable. Further, an explanation as to why these aren’t found (for example maybe they weren’t preserved or maybe they weren’t present) would be highly useful.

3) In addition, a conclusion of the present study (lines 189-191) is that, within cnidaria, medusae are primitive, and then the transition to a polyp stage occurs. I don’t want to say this is impossible, but this isn’t what is usually assumed. Instead, most would argue polyp evolved first (because of complete absence of medusae in Anthozoa, because the polyp develops before the medusae in those forms that possess it, etc.). So, the authors should spell out their conclusions on this in greater detail and state that they are suggesting a different character polarity for this key feature relative to most other studies.

---

## [Author Response]

Reviewer #2:Using scanning electron microscopy, the authors of this paper have documented some of the earliest evidence of relic muscle tissue in the fossil record. Their interpretations of the histology of their fossil material is based on detailed comparisons of exceptionally preserved relic soft parts in Olivooides, an extremely important early Cambrian genus, with putatively homologous tissues in extant organisms. Their work seems thorough and well-executed, and their interpretations, both of the documented fossil material and of the broader implications of their findings, seem reasonable and well supported by the available evidence. No doubt this study will stimulate further discussion and analysis of the fundamental empirical and theoretical problems in play here, including the timing and consequences of key anatomical innovations in the animal kingdom.A fascinating study, beautifully illustrated and cogently argued. My only suggestion for revision/improvement is the following: Near the beginning of the introductory paragraph starting with line 60, cite the study of Van Iten et al., (2016) on key cnidarian and putative cnidarian macrofossils (including Haaotia) from the Ediacaran of North and South America (you know the paper). Of particular relevance is Paraconularia sp. from the terminal Ediacaran Tamengo Formation. This fossil constitutes by the far the strongest fossil evidence of the presence of medusozoans (conulariids) during Ediacaran times. So, while the origin of the fine wrinkles in Haootia may be uncertain (and I would go a bit easy on this hypothesis, especially as you seem to be making a similar case for a putative early Cambrian "polyp" or sessile medusa), there is little doubt that medusozoans are present in the Ediacaran fossil record. Moreover, and by analogy with extant coronate polyps (which you illustrate), conulariids presumably had well developed longitudinal and circular muscles.

Many thanks for your suggestion concerning conulariids. We added the following sentences (lines 68-75 in the clean version; lines 69-76 in green in the marked version) and corresponding references:

“Conulariids is an extinct group of marine animals characterized by a hard exoskeleton resembling an ice-cream cone with typically a tetraradial symmetry, grooved corners and a periderm with numerous transverse ribs. Conulariids have been resolved as stem-group Scyphozoa (Van Iten et al., 2006) and have very likely Precambrian ancestors such as *Vendoconularia triradiata* and *Paraconularia* sp. from the terminal Ediacaran of Russia (Van Iten et al., 2005; 2016; Ivantsov, 2017) and Brazil (Van Iten et al., 2014; 2016), respectively. *Corumbella werneri* also supports a Precambrian origin of cnidarians, its annulated tube with a square cross-section and a lamellar microfabric resembling that of conulariids (Pacheco et al., 2015).”

We deleted the following sentence concerning *Hootia: “*Whether this external network corresponds to underlying muscles is uncertain and seems at odd with the assumed sessile lifestyle of the animal.*”*

Reviewer #3:The strengths of the paper are that it very nicely reviews the literature on the early fossil record of medusozoans, it gives a good summary and discussion of previous finds, it provides comprehensive and high quality pictures of the fossils with well labeled illustrations and interpretations, and addresses very important topics in paleontology, evolutionary biology, and invertebrate zoology.My suggested revisions follow. These involve requests for providing more details on a few matters that could perhaps be viewed as weaknesses, or at least I would judge that making the revisions I have suggested would strengthen aspects of what is already a very strong paper.1) It would be worth highlighting a bit more how some of the character information that conflicts with the view that olivooids are cnidarians. For instance, their pentaradial symmetry is very problematic. I think it is reasonable to interpret the structures the authors have identified as circular muscles but I think a bit more information as to how these fossils cannot possibly be echinoderms would be very valuable. Would muscles similar to the ones you found not be present in Cambrian (or modern) echinoderms? It would be worth stating or illustrating differences between the two in the revisions. That would provide some evidence supporting what you are arguing is the independent origins of pentaradial symmetry in two different groups.

Thank you for your interesting remark.

Indeed, pentamerous symmetry is a major feature of many adult echinoderms. However, both the vast majority of extant Palaeozoic members of the group have (1) a calcific skeleton formed of discrete plates with a mesh-like stereo structure, (2) a water vascular system with tube feet arranged along branches (ambulacra), and (3) a straight or coiled gut between mouth and anus. The pentamerous symmetry of adult echinoderms results from the metamorphosis of a bilaterian larva by the twisting of the body around the oral-aboral axis. None of these typical anatomical and developmental features is found in Cambrian olivooids. Moreover, no echinoderm has a muscle system comparable to that of olivooids (this paper). Some authors (e.g. Steiner et al., 2014) argued that the pentamery of olivooids would point to affinities with scalidophorans in which scalids and pharyngeal teeth display a clear 5-fold symmetry. However, extant and Cambrian scalidophoran have a through gut, a complex pharyngeal system and an eversible introvert that form early during their larval development. Again, none of these features can be observed in olivooids (both post-embryonic and polyp stages). Detailed analyses using X-ray microtomography have revealed that the internal organization of olivooids shared important features with that of cnidarians and especially medusozoans (see Han et al., 2013; 2016; Dong et al., 2013; 2016; Wang et al., 2017). Placing olivooids within cnidarians appears to be the most suitable position in the present state of knowledge. That some early cnidarians had an atypical pentamerous symmetry remains plausible although requiring explanation (see Gershwin, 1999). Although the vast majority of modern medusozoans are characterized by a tetraradial symmetry, several exceptions are worth being noted here (see Wang et al., 2017 for discussion). For example, *Eirene pentanemalis* is a hydrozoan (Leptomedusae, Eirenidae, Lin et al., 2013) with a remarkable pentamerous symmetry exemplified by five radial canals, five gonads near the middle part of the radial canals and five lip-like features around the mouth.

In summary, we don’t find it necessary to refute again the “echinoderm” and “scalidophoran” hypotheses since both lack strong support and been already reviewed (see Wang et al., 2017).

2) Also, the authors seem to be stating that no radial smooth muscles are found in their fossils. A more definitive statement of this would be valuable. Further, an explanation as to why these aren't found (for example maybe they weren't preserved or maybe they weren't present) would be highly useful.

The following sentence was added to the revised version (see lines 188-189 in the clean version; lines 193-194 in green in the marked version):

“However, no clear individual fibers could be distinguished in these adradial areas possibly owing to decay or taphonomy.”

We don’t reject the possibility that radial muscles were presented.

3) In addition, a conclusion of the present study (lines 189-191) is that, within cnidaria, medusae are primitive, and then the transition to a polyp stage occurs. I don't want to say this is impossible, but this isn't what is usually assumed. Instead, most would argue polyp evolved first (because of complete absence of medusae in Anthozoa, because the polyp develops before the medusae in those forms that possess it, etc.). So, the authors should spell out their conclusions on this in greater detail and state that they are suggesting a different character polarity for this key feature relative to most other studies.

This is also a valuable remark. What we observe in olivooids is a gradual change from an ovoid shape to a polyp-like conical structure. The ovoid stage shows an internal organization that strongly recalls that of a medusa with the oral side facing upwards (Wang et al., 2020). That’s why we assume that the development of olivooids occurred from a supposed medusoid form to a more mature polypoid form (and not the other way round). Indeed, the medusoid stage or ‘polyp-shaped medusa’ of olivooids differs from that of modern cnidarians where medusae have their oral side directed downwards and swim actively.